# Processed Animal Proteins from Insect and Poultry By-Products in a Fish Meal-Free Diet for Rainbow Trout: Impact on Intestinal Microbiota and Inflammatory Markers

**DOI:** 10.3390/ijms22115454

**Published:** 2021-05-21

**Authors:** Giulia Gaudioso, Giulia Marzorati, Filippo Faccenda, Tobias Weil, Fernando Lunelli, Gloriana Cardinaletti, Giovanna Marino, Ike Olivotto, Giuliana Parisi, Emilio Tibaldi, Kieran Michael Tuohy, Francesca Fava

**Affiliations:** 1Nutrition and Nutrigenomics Unit, Department of Food Quality and Nutrition, Research and Innovation Center, Fondazione Edmund Mach, 38098 Trento, Italy; giulia.gaudioso@fmach.it (G.G.); giulia.marzorati@fmach.it (G.M.); tobias.weil@fmach.it (T.W.); kieran.tuohy@fmach.it (K.M.T.); 2CIBIO—Department of Cellular, Computational and Integrative Biology, University of Trento, 38123 Trento, Italy; 3Aquaculture and Hydrobiology Unit, Technology Transfer Centre, Experiment and Technological Services Department, Fondazione Edmund Mach, 38098 Trento, Italy; filippo.faccenda@fmach.it (F.F.); fernandolunelli@gmail.com (F.L.); 4Department of Agricultural, Food, Environmental and Animal Science, University of Udine, 33100 Udine, Italy; gloriana.cardinaletti@uniud.it (G.C.); emilio.tibaldi@uniud.it (E.T.); 5Institute for Environmental Protection and Research (ISPRA), 00144 Roma, Italy; giovanna.marino@isprambiente.it; 6Department of Life and Environmental Sciences, Polytechnic University of Marche, 60131 Ancona, Italy; i.olivotto@univpm.it; 7Department of Agriculture, Food, Environment and Forestry, University of Florence, 50144 Florence, Italy; giuliana.parisi@unifi.it

**Keywords:** rainbow trout, aquaculture, sustainability, gut microbiota, inflammation, *Hermetia illucens*, poultry by-products, feed-borne microbiota

## Abstract

Sustainability of aquaculture is tied to the origin of feed ingredients. In search of sustainable fish meal-free formulations for rainbow trout, we evaluated the effect of *Hermetia illucens* meal (H) and poultry by-product meal (P), singly (10, 30, and 60% of either H or P) or in combination (10% H + 50% P, H10P50), as partial replacement of vegetable protein (VM) on gut microbiota (GM), inflammatory, and immune biomarkers. Fish fed the mixture H10P50 had the best growth performance. H, P, and especially the combination H10P50 partially restored α-diversity that was negatively affected by VM. Diets did not differ in the Firmicutes:Proteobacteria ratio, although the relative abundance of Gammaproteobacteria was reduced in H and was higher in P and in the fishmeal control. H had higher relative abundance of chitin-degrading *Actinomyces* and *Bacillus*, *Dorea,* and *Enterococcus*. *Actinomyces* was also higher in H feed, suggesting feed-chain microbiome transmission. P increased the relative abundance of protein degraders *Paeniclostridium* and Bacteroidales. IL-1β, IL-10, TGF-β, COX-2, and TCR-β gene expression in the midgut and head kidney and plasma lipopolysaccharide (LPS) revealed that the diets did not compromise the gut barrier function or induce inflammation. H, P, and H10P50 therefore appear valid protein sources in fishmeal-free aquafeeds.

## 1. Introduction

Fish is recommended as a nutritious source of dietary protein for human health, and farmed fish have great potential for environmental sustainability according to the EAT Lancet Commission on Healthy Diets from Sustainable Food Systems [1]. Aquaculture is also currently the fastest growing food production sector, surpassing wild caught fish in terms of global annual production [1,2]. In order to decrease the future environmental footprint of aquaculture, several studies are evaluating alternative and more sustainable sources of feed protein-rich ingredients, able to ensure fish health while retaining nutritional and organoleptic quality for the consumer.

In the past, searching for high-performance fishmeal (FM)-free formulations led to the development of innovative aquafeeds comprising new sustainable alternative ingredients. Vegetable protein-rich feeds (VM) have been used as sustainable alternatives to high-performance fishmeal (FM) [3], although VM often suffer from inferior growth performance [4] and changes in immune function [5]. Endogenous anti-nutritional factors and complex indigestible carbohydrates present in VM, together with low levels of essential amino acids and insufficient n-3 PUFA, lead to adverse effects on fish health and the nutritional quality of the final product. Moreover, intestinal barrier integrity as a consequence of VM diets leads to impaired nutrient absorption, alterations in the gut-associated immune system, and changes in resident gut microbiota composition [5]. Alternative animal-based protein sources therefore may offer considerable advantages over plant-derived protein. Poultry by-product meal (P) is a candidate animal-sourced alternative protein source that was re-authorized for use by the European Union in 2013 [6]. P is obtained from rendered and clean by-products of the poultry processing industry; it has an energy content similar to that of FM, with a well-balanced amino acid profile and good palatability, which are key attributes for carnivorous fish diets [7]. In addition, animal by-products have a lower carbon footprint compared to FM or vegetable alternatives [3,8]. Previous studies showed that replacement of high amounts of dietary proteins from FM or VM with P gave good results in terms of growth performance in salmonids [9,10,11]. However few studies have examined the growth performance and animal health effects of increasing percentages of P in FM-free, VM-based feeds in rainbow trout (*Oncorhynchus mykiss*) [12]. Insect meal (IM) also represents a valid alternative protein source to VM and possibly FM for aquaculture formulations. Compared to poultry by-products, insects have high/medium protein levels. In addition, certain insect meals have been shown to contain biologically active compounds, such as chitin, antimicrobial peptides, and short–medium fatty acids (FAs) [13], which have been associated with improved fish innate immune function (promoting immunomodulatory effects) and modulation of gut microbiome composition [14,15,16]. Interestingly, employment of *Hermetia illucens* meal (H) at a low percentage in fish feed resulted in immunostimulation (up-regulation of genes encoding for IL-1β, IL-17F, and TNF- α), probably due to its chitin, antimicrobial peptides, and fatty acids content. Immunomodulatory and antimicrobial effects of H seem to be dependent on both the insect species and the diet of the insect [17]. Insects also have the advantage of growing on a wide range of substrates, and it is possible to improve their nutritional profile simply by changing their growth substrate. They have also a low-environmental impact since insects can grow on waste derived from the vegetable food industry [18]. 

Fish health deeply influences growth performance, and, in turn, fish wellbeing is influenced by gut health. Absorption of nutrients, immune function, and the inflammatory state are strongly linked to gut health and the composition of the resident gut microbiota. Previous studies highlighted how the gut microbiota (GM) and the diet are strictly interconnected and influence the inflammatory, immune, and nutritional status of fish [19,20]. GM could mediate the host immune response by stimulating the production of several soluble mediators of inflammation and immune cell recruitment. Fish GM plays a crucial role in gut-associated lymphoid tissue (GALT) maturation and development, also protecting the fish from infection [21,22,23]. The gut microbiota and its metabolites (such as short-chain fatty acids, SCFAs) are directly involved in maintaining intestinal barrier function and integrity, thus supporting fish growth rates [24]. The composition of the gut microbiota and relative abundances of different members of the gut microbiota respond to the nutritional composition and ingredients used in aquafeeds [25].

Rainbow trout is the most commonly farmed teleost fish species in Italy, with 36.800 tons produced in 2019 [26]. Its gut microbial community is mainly dominated by Actinobacteria, Firmicutes, and Proteobacteria, whose proportions are strongly influenced by dietary proteins. A higher Firmicutes:Proteobacteria ratio seems to be promoted by vegetable ingredients, particularly soybean proteins but also pea and canola proteins [27]. The effect of partial replacement of FM with a mixture of animal by-products and plant proteins on gut microbial populations of rainbow trout has been recently studied [28]. However, while Rimoldi and colleagues employed a mixture of P, animal proteins, including swine blood, and vegetable proteins to substitute part of FM, and soybean oil to replace fish oil, in this current study, we examine the effect of adding different percentages of P to a VM-based FM-free diet. Replacement of fish proteins with alternative animal and vegetable proteins was previously shown to significantly change the abundance of Fusobacteriaceae, Bacteroidetes phyla, and several families and genera within the Proteobacteria and Firmicutes phyla, with variability between diets [28]. However, there is lack of knowledge on the impact of P in sustainable, FM-free, vegetable formulations on the gut microbiota. Insect meal inclusion (up to 30%) in rainbow trout feed was also shown to significantly affect dominant gut bacterial phyla such as Firmicutes, Tenericutes, Actinobacteria, and Proteobacteria and to promote the growth of *Mycoplasma* and Actinomycetales genera, specifically *Actinomyces* and *Corynebacterium*, as well as *Enterococcus*, *Lactobacillus,* and *Staphylococcus*, whilst decreasing *Bacillaceae*, especially *Bacillus* and *Oceanobacillus* genera [16]. The magnitude of differences in bacterial abundance reflected the percentage of insect proteins in the diet [15]. Insect-derived ingredients also increased the abundance of lactic acid bacteria within the Firmicutes phylum in trout intestinal contents [14]. There is some evidence from other fish species that a combination of H and P might promote fish growth and immune defense against pathogens, although no studies in trout have investigated the mixture H + P in rainbow trout microbiota in an FM-free vegetable aquafeed [29].

In this study, which is part of a larger investigation [30], we studied the effects of including a commercial insect meal (H) from *Hermetia illucens*, poultry by-product meal (P), and their combination (H + P) on the gut microbiota composition and biomarkers of inflammation and gut barrier integrity in rainbow trout fed a fish meal-free plant-based diet.

## 2. Results

### 2.1. Growth Parameters

The fish readily accepted the experimental diets, and all feeds were consumed without rejection or loss. In all dietary treatments, the average body weight of fish fed over three months (69 meals) more than tripled at the end of the feeding trials, and the survival rate of the fish was 100% for all the experimental groups. Fish fed the diet coined H10P50 showed a higher final body weight than fish fed either CV or CF (*p* < 0.001), and their weight did not differ (*p* > 0.05) from the other dietary treatments (Table 1). In addition, fish fed H10P50 showed a higher total length compared to fish fed CF or CV diets. The corresponding Fulton’s condition factor (K) was found to indicate well-proportioned growth under all experimental conditions. CV showed higher K values than CF and comparable K to all H groups. Insect meal groups had significantly higher K than P groups, while the mixed H + P meal had a significantly higher K factor than both P groups (Table 1). All other zootechnical parameters, such as the specific growth rate (SGR), feed intake (FI), and feed conversion ratio (FCR), were already reported and discussed elsewhere by Randazzo and colleagues [30].

### 2.2. Gut Microbial Ecology

Illumina MiSeq sequencing of gut microbial 16S rRNA gene amplicons produced a total of 8,798,863 reads, with 129,395 ± 28,897.18 raw reads per sample. Raw sequences were submitted to the European Nucleotide Archive (PRJEB43152). After QIIME2 analysis, we removed sequences that were identified as Cyanobacteria, and the total number of reads was 2,023,741, with an average of 29,760.89 ± 30,730.25 reads per sample. Fish feed was also sequenced and generated a total of 3,960,358.98 raw reads, with 141,441.39 ± 38,974.52 reads per sample. After QIIME2 quality filtering and removal of reads identified as Cyanobacteria, the total number of reads of fish feeds was 2,018,291, with 29,680.75 ± 30,762.37 reads per sample.

The fish feed composition had a strong impact on the microbial composition in terms of α-diversity indexes, particularly showing that vegetable-based diets drastically reduced microbiota complexity. CF had the highest bacterial richness in terms of observed features and the Chao1 index of all the experimental groups (Figure 1A,B). On the other hand, CV showed the lowest values for both indexes (observed OTUs 73.67 ± 28.97 and Chao1 77 ± 32.30). When insect and poultry meal ingredients were introduced in the feed, α-diversity was partially restored, and the difference with CF tapered. Amongst tested feeds, dietary inclusion of 23% and 45% of *Hermetia illucens* (H30 and H60, respectively) did not differ in bacterial α-diversity compared to CF (*p* = 0.565 and *p* = 0.585, H30 vs. CF; *p* = 0.511 and *p* = 0.503, H60 vs. CF, Chao1, and number of features, respectively). In addition, the community evenness of the feed containing insect meal was similar to that of fishmeal, although the difference in the Shannon index did not reach statistical significance (Figure 1C). β-diversity (Bray Curtis dissimilarity, weighted and unweighted Unifrac analysis) showed a clear separation between diets, with H diets and P diets clustering together (Figure 1D).

The results of gut microbiota analysis and the significant differences in relative abundance of taxa are illustrated in Figure 2 and Table 2.

Proteobacteria, Firmicutes, and Tenericutes were the most represented phyla in all the fish, regardless of diet, covering between 84% and 98% of all identified phyla (Figure 2A). No significant differences were observed in the relative abundances of Proteobacteria, Firmicutes or Tenericutes between diets. In addition, no differences (*p* > 0.05) were observed in the Firmicutes:Proteobacteria ratio between the dietary groups. We observed fewer Gammaproteobacteria in all diets including the insect meal (H) compared to CV, with lower abundance in those diet groups with a high percentage of *Hermetia* in the feed. Gammaproteobacteria abundance in the P30 and P60 dietary groups was not different (*p* > 0.05) from that of CF (Table 2A). Actinobacteria were a less dominant member of the gut microbiota of all trout, mainly colonizing *Hermetia-*fed animals, where they constituted 3–6% of all phyla. The Actinobacteria phylum was affected by the concentration of the *Hermetia* meal in a dose-dependent manner, with significantly higher abundance in H60 compared to P60 (Figure 2A and Table 2, *p* = 0.037). Bacteroidetes were less prevalent, with an average relative abundance of 0.6 ± 0.4% across all samples. Bacteroidetes were significantly higher in the P30 group compared to H60 (1.35 ± 1.26% vs. 0.09 ± 0.09%, P30 vs. H60 relative abundance respectively; *p* = 0.016, Table 2 and Appendix A). Fusobacteria were only observed in CF, with a noteworthy percentage of relative abundance, but also with high inter-individual variability (13.5 ± 29.91%; *p* < 0.05). This was due exclusively to the presence of the *Cetobacterium* genus, which was a dominant member of the microbiota in three specimens fed CF (13.44 ± 29.91%). *Weissella*, *Enterococcus*, *Lactobacillus*, *Leuconostoc*, *Streptococcus,* and *Lactococcus* of the Lactobacillales order were the most represented genera within the Firmicutes phylum, regardless of diet. The *Weissella* genus was present in all animals, with the highest relative abundance in the CF and H10P50 group and the lowest in CV (Figure 2B). Within Lactobacillales, the *Enterococcus* genus was found at significantly higher relative abundance in *Hermetia* diets (Table 2). In addition, *Dorea* was mainly present in H-fed trout, with significantly higher relative abundance in H10, H30, H60, and H10P50 compared to CV and to P60 (Figure 2B, Table 2; *p* < 0.05, Appendix A). All insect-fed trout had higher intestinal relative abundance of the *Actinomyces* genus compared to CV and to both P groups (Figure 2B, Table 2; *p* < 0.05, Appendix A). In addition, the group fed the insect and poultry protein mixture (H10P50) had higher intestinal levels of *Actinomyces* than CV animals (Figure 2B; *p* = 0.04). Trout fed increasing percentages of *Hermetia* meal had progressively lower relative abundance of *Pseudomonas,* and this was statistically significant for H60 compared to CV (H60: 1.53 ± 1.89%, CV: 9.80 ± 11.06%; *p* = 0.04, Figure 2B, Table 2). An opposite trend was observed for the *Bacillus* genus, which was significantly higher in H60 compared to CV (H60: 3.60 ± 3.09%, CV: 0.07 ± 0.14%, relative abundance mean ± SD; *p* = 0.036). Similarly, *Oceanobacillus* was mainly recovered from intestinal contents of H-fed trout, and its relative abundance was significantly higher than that in CV and P, but not compared to CF (Table 2).

The *Erysipelothrix* genus was almost absent in animals fed CV or P, while its relative abundance increasing with an increasing percentage of *Hermetia* meal in the diet. A similar trend was observed for *Pseudoramibacter Eubacterium*, a member of the *Eubacteriaceae* family, within the Firmicutes phylum, which was present in *Hermetia* groups but not in Ps and CV (Figure 2A,B, Appendix A).

### 2.3. Microbiota Composition of Fish Feed

β-diversity showed a good separation of microbiota isolated from feed pellets (*p* < 0.001; Figure 3C), but no statistically significant differences were observed in relative abundances, probably due to the fact that only three replicates of feed pellet for each diet were sequenced. However, 16S rRNA sequencing of fish feed pellets revealed that Bacteroidetes, Firmicutes, and Proteobacteria were the dominant phyla in all the experimental feeds (Figure 3C). The relative abundance of Actinobacteria increased with increasing percentage of insect meal from 2.27 ± 0.44% in H10 feed to 3.38% ± 0.81% in H30 and 5.94 ± 1.14% in H60. Within the Actinobacteria, the *Actinomyces* genus was mainly present in insect meal-enriched feeds (H10: 1.12 ± 0.26%; H30: 2.03 ± 0.57%; H60: 3.50 ± 0.72%) (Figure 3C). The relative abundance of Bacteroidetes in the feed increased proportionally with the insect meal content (3.36 ± 0.27% in H10, 6.04 ± 0.93% in H30, 9.77 ± 1.22% in H60, 3.16 ± 0.29% in H10P50), while it was present at a much lower relative abundance in the other feeds (0.38 ± 0.27% in CF, 1.16 ± 1.00% in CV, 0.61 ± 0.31% in P30, 1.45% ± 1.18% in P60) (Figure 3A). This was mainly due to a higher presence of genera *Bacteroides* (1.43 ± 0.18% in H10, 2.28 ± 0.19% in H30, 4.05 ± 0.37% in H60) and *Dysgonomonas* (0.47 ± 0.31% in H10, 1.83 ± 0.43% in H30, 2.64 ± 0.24% in H60) (Figure 3C). An opposite trend was observed for the Proteobacteria phylum, which decreased both with increasing percentage of *Hermetia* meal (65.45 ± 1.08% in H10, 61.06 ± 1.57% in H30, 52.33 ± 1.55% in H60) and of the poultry by-product meal (70.65 ± 1.49% in P30, 68.62 ± 1.76% in P60, and 64.12 ± 0.60% in H10P50). Within Firmicutes, *Lactobacillus* and *Weissella* represented the genera at the highest relative abundance in H and P feeds (Figure 3C). *Weissella* relative abundance decreased with increasing quantity of insect meal in the feed (9.90 ± 2.21% in H30, 7.25 ± 0.69% in H60), and it was also higher in CF compared to CV (12.91 ± 2.63% vs. 10.66 ± 3.52%, respectively). The highest relative abundance of Lactobacillus was found in CF (11.43 ± 0.95%), while the lowest in H60 (5.78 ± 0.69%). Proteobacteria was the most abundant phylum in CF feed, mainly represented by the Rickettsiales order (62.08 ± 3.09%). Within Proteobacteria, CF had the highest relative abundance of the *Photobacterium* genus (4.07 ± 1.19%). Conversely, this genus decreased in concomitance with a higher content of *Hermetia* in feeds (0.84 ± 1.29% in H10, 0.56 ± 0.85% in H30 and 0.30 ± 0.49% in H60).

### 2.4. Correlation Analysis between GM and Ingredients

Spearman’s correlation analysis was performed to correlate microbial relative abundances with dietary ingredients. Statistically significant differences were observed between GM taxa and dietary ingredients (Figure 4). *Hermetia* meal showed a strong positive correlation with *Actinomyces*, *Bacillus*, *Corynebacterium*, *Dorea*, *Enterococcus*, *Erysipelothrix*, *Oceanobacillus*, *Paenibacillus*, and *Pseudoramibacter Eubacterium* (*p* < 0.001 for all correlations). An opposite trend was observed between the L-Tryptophan concentration and the same genera (*p* < 0.01). The presence of poultry by-product meal showed a significant (*p* < 0.05) negative correlation with *Actinomyces*, *Erysipelothrix*, *Enterococcus*, *Pseudoramibacter Eubacterium* and with *Oceanobacillus*. P was positively correlated with *Paeniclostridium* within the *Peptostreptococcaceae* family (*p* < 0.01), while genus *Peptostreptococcus* showed a significant positive correlation with fish meal (*p* < 0.05). No other statistically significant differences were observed when correlating bacterial taxa with growth parameters, gene expression or plasma lipopolysaccharide (LPS) concentration results (Appendix A.)

### 2.5. Gene Expression in the Midgut and Head Kidney

The midgut and head kidney showed different expression patterns of selected markers. Significantly higher levels of the pro-inflammatory cytokine IL-1β were observed in head kidney biopsies of fish fed with H10, H30, H60, and P30 compared with CV (H10 vs. CV, *p* = 0.016; H30 vs. CV, *p* = 0.016; H60 vs. CV, *p* = 0.016; P30 vs. CV, *p* = 0.016) (Figure 5B). An opposite trend was observed in midgut biopsies, with lower IL-1β in *Hermetia* diets compared to CV, although not statistically significant (Figure 5A). A similar pattern for TGF-β was found in both midgut and head kidney biopsies, with higher levels of expression in H60, P30, and H10P50 than in CV, even if not statistically significant (Figure 5E,F). No significant changes were observed in IL-10 (Figure 5C,D), COX-2 (Figure 5G,H), and TCR-β (Figure 5I,J) among dietary groups.

### 2.6. Plasma LPS as an Intestinal Permeability Marker

No significant differences in LPS were observed between experimental groups (Table 3).

## 3. Discussion

Replacement of fishmeal with vegetable plant protein meal was previously shown to affect the growth and general wellbeing of carnivorous fish [27]. The gut microbiota is strongly involved in digestion and absorption of nutrients and is markedly modulated by dietary composition [31,32]. In the search for sustainable protein sources to improve VM-based FM-free formulations, this study investigated the effect of including different percentages of *Hermetia illucens* meal (H) and poultry by-product meal (P) on gut microbial composition, as well as gut and systemic biomarkers of inflammation.

Overall, at the end of the feeding trial, fish showed a good performance in response to the test diets containing H or P meal, both singly and in combination, when compared to the vegetable or fish meal control formulations, as already reported and discussed elsewhere by Randazzo and colleagues (2021) [30]. The lower growth rate observed in trout fed the CV diet is in line with a previous study performed on the same fish species with similar size [33], where the FM substitution that ranged from 75% to 100% with the vegetable counterpart negatively affected the fish growth performance by up-regulating the gene involved in the white muscle lysosomal proteolysis, thus affecting its growth. In the present study, the observed improvements in the growth and feed efficiency (data not shown) of trout fed the FM-free diet combining H and P (H10P50) compared to both controls diets (CV and CF) are not easily comparable to other studies on different or the same fish species. In fact, in previous studies, *Hermetia illucens* or poultry by-product meals were mainly evaluated as FM substitutes in diets containing variable, but low levels of vegetable proteins [10,34,35]; while in the present study, the alternative ingredients were used to replace vegetable protein meals from the CV diet preparation lacking fish meal. Since all diets have been formulated to fit the rainbow trout nutrient requirements, the better final body weight and feed conversion efficiency observed mostly in all FM-free diets could possibly result from either a better overall digestible amino acid balance or improved gut health and nutrient digestibility, or even both. Fulton’s condition factor is an easy key performance index used to compare the condition, fatness, or wellbeing of fish, based on the assumption that heavier fish of a given length are in better condition [36]; K values less than 1 imply that fish are not in good state of well-being within their habitat, while values greater than 1 imply that fish are in good physiological state of well-being. The Fulton’s K values reported in the present study, independently from the dietary treatments, were all greater than 1 and were similar to those recorded in rainbow trout with a comparable final size fed a plant protein mixture diet [37].

The detrimental effect of high VM diets in salmonids has been widely demonstrated [27,38,39]. Anti-nutritional factors (ANF) are thought to be the main culprits inducing mucosal inflammation, disruption of the intestinal barrier, inhibition of digestive enzymes, and/or reduced nutrient uptake [38]. Few studies in rainbow trout have employed exclusively vegetable diets and also monitored changes in GM. Modulation of the gut microbiota may be exploited to potentially taper the observed undesirable effects of VM. Inclusion of ingredients such as P and H might potentially act in synergy with gut bacteria to stimulate gut functions and reduce inflammation. In fact recent studies have highlighted increased feed efficiency parameters when poultry by-product meal or insect meal were used to partially replace FM [28,40], but fewer studies have investigated the impact of P or H in a totally FM-free vegetable formulation. Here, we analyzed the specific impact of P and/or H as sole sources of animal protein replacing vegetable protein with increasing doses in a FM-free diet (CV).

Gut microbiota analysis through 16S rRNA sequencing revealed that the test diets induced significant differences in intestinal bacterial profiles. A marked reduction of gut microbiota α-diversity in terms of species richness and evenness was observed for all the vegetable-based diets compared to the fishmeal diet (CF). High gut microbial richness and biodiversity were previously suggested as gut health indicators since low-diversity or alterations of gut microbial ecosystems have often been related to acute bacterial infections in rainbow trout [41]. The analysis of α-diversity showed that the GM of trout fed CV had the lowest bacterial richness, both in terms of observed features and the Chao1 index, while animals fed CF had the highest bacterial richness (Figure 1A,B). Interestingly, the inclusion of different percentages of *Hermetia* meal positively influenced the alpha diversity and partly restored the loss of bacterial diversity in fishmeal-free diets, thus reaching values close to those obtained with CF. Insect meal also influenced community evenness, even if no significant differences were observed using the Shannon index (Figure 1C). Notably, these results are in agreement with recent studies where bacterial richness and evenness increased with increasing percentages of *Hermetia* meal in the diet [14,40,42]. Chitin was previously suggested as the major driver of this increase of alpha-diversity. Chitin is a structural polysaccharide and a primary component of the insect exoskeleton that behaves as an insoluble fiber since it reaches the distal part of the intestine almost undigested. Fermentation of chitin and its derivative chitosan leads to the production of several beneficial compounds, such as short-chain fatty acids (SCFAs), in particular acetate, a precursor of the synthesis of butyrate, which exerts a well-known positive effect on gut and overall fish health, nutrient utilization, and disease resistance [43,44,45,46,47,48]. Several species of *Lactobacillus* and other non-lactic bacteria such as *Bacillus* have been used as chitin fermenters [49], and this suggests that they may have a similar role in the fish gut. In our study, the chitinase producer *Bacillus* genus was found at high concentration in trout fed the highest H feed, as discussed later. Chitin may therefore act as a prebiotic by supporting the growth of beneficial chitin degraders, thus increasing gut bacterial richness and biodiversity and possibly stimulating intestinal fermentation.

The gut microbial community of all experimental groups was dominated by Tenericutes, Proteobacteria, Firmicutes, Actinobacteria, Bacteroidetes, and Fusobacteria phyla, regardless of diet. These results are in line with previous analysis of rainbow trout GM [28,41]. Tenericutes represented the most abundant phylum in trout microbiota, followed by Proteobacteria and Firmicutes. Previous analysis of trout GM in response to FM substitution with vegetable proteins or with *Hermetia* meal highlighted a shift in the Firmicutes:Proteobacteria ratio due to lower Proteobacteria in the presence of high VM or high H in the feed [27]. However, in our feeding trial we did not observe significant differences in the overall abundance of Firmicutes or Proteobacteria or the Firmicutes:Proteobacteria ratio between the diets. Our results might be explained by the high quality of the vegetable protein used in our diets (Appendix A). As already suggested, purified and processed plant proteins might induce smaller impairments of gut microbiota, probably due to lower ANF and increased protein availability [27]. An increase in Firmicutes was previously observed after feeding fish with diets including insect meal [14], and we also found slightly higher Firmicutes in H groups, although this was not significant compared to other diets (Table 3). The aundance of Proteobacteria in *Hermetia-*fed animals was slightly lower than that in CF and CV, while in poultry by-product meal, the dietary abundance of Proteobacteria was more similar to both CF and CV (Figure 2A). In particular, Gammaproteobacteria of trout fed the highest *Hermetia* concentration were significantly lower than in CV, while all the other diets had comparable levels of Gammaproteobacteria between each other and similar to CF (Table 3, Appendix A). Within Gammaproteobacteria we did not detect any potential pathogens, such as Vibrionales or Aeromonadales, different from previous reports [28]. Within Gammaproteobacteria, the *Pseudomonas* genus was a dominant member. *Pseudomonas* was significantly lower in H60 compared to CV. The highest relative abundance of *Pseudomonas* was found in CV, while H30- and all P-fed animals harboured similar levels to CF. *Pseudomonas*, together with *Bacillus*, *Serratia*, *Cetobacterium,* and *Lactobacillus* are common dominant members of the gut microbiota of healthy farmed trout [41]. High abundance of Rickettsiales within Alphaproteobacteria was also found in all the samples, ranging between 17% and 43% of all identified orders and apparently not affected by diet. This intracellular bacterium has been seen to have both a symbiotic as well as a pathogenic relationship with its host and it appears not to be necessarily linked to disease, similar to what we observed in this study [50].

Fish fed H10, H30, and H60 showed an increase in the relative abundance of bacteria belonging to the *Mycoplasma* genus, within the Tenericutes phylum, although not significant after FDR. This observation agrees with previous studies, where *Mycoplasma* was found as one of the most abundant genera in rainbow trout and Atlantic salmon intestines [40,42,51]. *Mycoplasma* uses dietary substrates to produce lactic acid and acetic acid as the main fermentation end-products [52]. Moreover, a decrease in *Mycoplasma* abundance in rainbow trout intestine has recently been related to a higher susceptibility to disease and inflammation [53]. *Mycoplasma* was previously seen to colonize the GM of farmed salmonids and to be less affected by diet. Nevertheless, it has an essential role in maintaining gut homeostasis in rainbow trout.

We observed a significant increase of Actinobacteria in the GM of animals fed different percentages of *Hermetia* meal in the diet. The abundance of Actinobacteria ranged from about 3% in H10 to 5% in H30 and reached 6% in H60, while the same animals showed a proportional decrease in Proteobacteria, with 55% in H10, 24% in H30, and 20% in H60 (Figure 2A). This is consistent with other studies where a similar shift in these two dominant phyla was observed with a partial substitution of fish meal with insect meal [16,42]. Our results showed that all insect meal-including diets had significantly higher relative abundances of the *Actinomyces* genus, within the Actinobacteria phylum, compared to CV (H10: 1.58%, H30: 3.16%, H60: 4.66%, H10P50: 0.52%, CV: 0.05%; Table 3). Bacteria belonging to this genus were previously identified as active chitin degraders, which were seen to increase fish growth and efficiency proportionally to the quantity of chitin in the diet [43]. The chitinolytic activity of *Actinomyces* leads to the production of chitosan, a partial-deacetylated derivative of chitin [54]. In support of this hypothesis, a positive Spearman’s correlation was observed between *Actinomyces* and insect meal (Figure 4). Another chitinase-producing genus, *Bacillus,* within the Firmicutes phylum, was significantly increased in relative abundance in H-fed trout. *Bacillus* is commonly used as a probiotic in aquaculture industries and is reported as one of the core taxa in the fish gut microbiome [55,56,57]. Several recent studies reported an increase in *Bacillus* in the GM of fish fed insect meal [14,40,42] and its increased relative abundance in the current study is probably related to the chitin content of the H diets (Appendix A). In support of this, we observed a strong positive Spearman correlation between *Bacillus* and insect meal. Microbiota analysis of feed pellets revealed that feed formulations containing *Hermetia illucens* carry both *Actinomyces* sequences and *Bacillus* sequences (Figure 3). These results suggest that on the one hand, the H diet may promote the growth of specific autochthonous bacteria that constitute the natural gut microbial community, and on the other, that such diets may also contribute to enrich allochthonous bacteria via microbiota transfer from insect feed to the fish gut, possibly impacting feed conversion and animal health. Microbiota transfer from feed has been previously reported. Li et al. (2021) [42] suggested that the increased *Actinomyces* relative abundance in rainbow trout fed black soldier fly may be explained by feed-borne microbiota. Similarly, in our study, *Photobacterium* had the highest abundance in CF-fed trout, while it was nearly absent in all the other experimental groups. A similar distribution was observed in the microbial composition of the feed, with *Photobacterium* being higher in CF than in all the other feeds. In accordance with a previous study [58], it appears that both *Actinomyces* and *Photobacterium* may transit from feed to the gut. The fish gut microbiota therefore behaves as a dynamic ecosystem which appears to be modelled by feed-associated bacteria and nutrient composition providing the host with acquired digestive functions [59]. Microbial transfer from feed to fish could therefore be exploited as a strategy to enhance fish capability to absorb nutrients.

Fish fed H10, H30, and H60 showed higher relative abundance of the *Erysipelothrix* genus. Moreover, Spearman’s correlation analysis showed a positive correlation between *Erysipelothrix* and the presence of insect meal in the diet. Rimoldi and collaborators (2021) [40] recently observed higher abundance of *Erysipelotrichaceae*, and especially *Erysipelothrix*, in the GM of trout fed an H-enriched diet. High relative abundance of *Erysipelothrix* was also previously found in animals fed high poultry by-product proteins (55%) in addition to FM, together with high abundance of *Enterococcaceae*, *Streptococcaceae,* and *Enterobacteriaceae* [28]. On the other hand, in our study, *Erysipelothrix* was almost absent in trout fed P and CV. Although the role of this genus in the fish intestine is not fully understood, some species belonging to *Erysipelothrix* such as *E. piscisicarius* cause systemic disease in several fish with necrotizing dermatitis and orofacial ulceration [60]. Further studies are required to fully understand the role of *Erysipelothrix* in rainbow trout metabolism and health. In addition, we reported a higher abundance of the *Enterococcus* genus, within *Enterococcaceae*, in H diets compared to the other diet groups. This is in contrast with Rimoldi et al. (2018) [28], who reported higher abundance of *Enterococcaceae* and, specifically, of the *Enterococcus* genus, together with high values of *Eryspelothrix* after feeding a high poultry by-product and animal protein commercial diet. However, in the study by Rimoldi et al. (2018) [28], the diet that showed the highest abundance of *Enterococcus* is not comparable to any of our P diets, due to a much higher animal protein content from various sources (poultry, swine) as well as FM content. In our study, *Enterococcus* abundance in H diets was much higher than that observed by Rimoldi and colleagues (2018) [28], while *Erysipelothrix*, on the other hand, had a much lower relative abundance.

*Dorea* genus, within the *Lachnospiraceae* family, was significantly higher in all H diets, including H10P50, compared to CV and to P60. The *Paeniclostridium* genus was significantly higher in H10P50 compared to CV (Table 3). Interestingly, two different genera within the *Peptostreptococcaceae* family correlated positively with fishmeal and with poultry meal, respectively (Figure 4). *Peptostreptococcaceae* was reported as one of the dominant members of the GM after P feeding in salmonids and, specifically, in rainbow trout [28,61]. *Peptostreptococcaceae* and *Lachnospiraceae* are fast protein degraders; therefore, distinctive microbial genera within the same family might be specialized in the digestion of different protein structures present in FM and P [62]. Both *Lachnospiraceae* and *Peptostreptococcaceae* were previously reported as indicator taxa for fast-growing fish [63]. Considering the good growth performance we observed when feeding trout with a mixture of insect and poultry meal, there may be a role for *Lachnospiraceae* and *Peptostreptococcaceae* in helping digestion and promoting animal growth.

In addition to the characterization of the gut microbial community, we performed a gene expression analysis of immune and inflammatory biomarkers in midgut and head kidney tissues. Different patterns of expression of the cytokines IL-1β, IL-10, TGF-β, and the inflammatory markers COX-2 and TCR-β were observed in the analyzed tissue biopsies. IL-1β is a pro-inflammatory cytokine and plays a central role in generating and controlling the immune response in rainbow trout during infection and inflammatory processes [64,65]. The early activation of IL-1β stimulates the recruitment of other cytokines and the activation of lymphocytes and macrophages [66]. Our results showed an overexpression of IL-1β in the head kidney of fish fed P30 and H10P50 diets compared to CV. An opposite trend of expression, even if not statistically significant, was reported in midgut biopsies. The expressions of the others selected biomarkers (i.e., IL-10, TGF-β, COX-2, and TCR-β) were not affected by dietary treatment. Interestingly, the anti-inflammatory cytokine IL-10 had a pattern of expression resembling that of IL-1β. This was observed in all dietary treatments, both in midgut and head kidney tissues, even if not statistically significant. IL-10 has a crucial role in controlling autoimmune and inflammatory reactions in fish [67] and its expression is commonly induced together with pro-inflammatory cytokines [68]. According to these considerations, we presume that the expression of IL-10 could have a role in modulating the immune response in trout limiting excessive inflammatory reactions, as already hypothesized by Heinecke and Buchmann in 2013 [69]. Nevertheless, since no significant changes were reported in the expression of all the other selected markers and given that changes in IL-1β expression were only observed in the head kidney of two dietary groups, we can conclude that none of the experimental diets had a deleterious impact on the overall immune health of the animals. Plasma LPS analysis supported this result. Circulating LPS is a marker of intestinal barrier integrity, since increased gut permeability may induce translocation of inflammatory bacterial LPS from the lumen to the blood circulation [70]. In this study, no significant differences in LPS concentration were observed between diets, thus suggesting maintenance of intestinal barrier integrity in response to the experimental diets.

## 4. Materials and Methods

### 4.1. Animal Ethics Statement

All the experiments were performed following directive 2010/63/UE on the protection of animals used for scientific purposes and in line with Italian legislation and approved by the Ethics Committee of the Edmund Mach Foundation (n. 99F6E.0). The study protocol was authorized by the Italian Ministry of Health (530/2018-PR). In order to avoid animal suffering, fish were euthanized by immersion in a lethal dose solution of tricaine methanesulfonate (MS-222, 1 g L^−1^) before tissue sampling.

### 4.2. Fish, Experimental Diets, and Culture Conditions

A total number of 1200 juvenile rainbow trout (*Oncorhynchus mykiss*) with an average weight of 54.2 g (±9.94) were randomly distributed into 24 square fiberglass tanks (1.6 m^3^) and assigned to eight different dietary treatments for three months. The growth trial was performed at the indoor experimental facility at the Technology Transfer Center (CTT), Edmund Mach Foundation (FEM), San Michele all’Adige, TN, Italy, as reported in Randazzo et al., 2021 [30]. Briefly, eight iso-proteic (N × 6.25, 42% dry matter), iso-lipidic (24% DM) and iso-energetic (approximately 23 MJ/kg) diets were manufactured at SPAROS Lda. (Olhão, Portugal) by extrusion. A diet rich in vegetable protein derivatives (control vegetable, CV) was prepared to have a 10:90 and 20:80 fish-to-vegetable protein and lipid ratios, respectively. By contrast, a fish-based diet (control fish, CF) was formulated with opposite fish-to-vegetable protein and lipid ratios. Six more diets were obtained by replacing graded levels of protein (10, 30, and 60%) of diet CV, by protein from partially defatted *Hermetia illucens* pre-pupae meal (H10, H30, and H60) or poultry by-product meal (P30, P60), while maintaining the same vegetable-to-fish lipid ratio as in the CV diet. An additional diet that replaced 60% vegetable protein with a combination of 10% and 50% of protein from *Hermetia illucens* and poultry by-product meal (H10P50) respectively, was also formulated. The ingredient composition and proximate analysis of the test diets are shown as supplementary material (Appendix A). During the feeding trial, a visual inspection of tanks was carried out daily to check feeding behavior and mortalities. The rearing conditions were monitored every week to maintain optimal environment control for rainbow trout growth. The mean dissolved oxygen (DO), measured after the daily meal in the tank water outlet, was 9.43 ± 0.42 mg/L, and the water temperature fluctuated close to 13.3 ± 0.23 °C. Fish were fed by hand twice a day, six days a week, at apparent visual satiety over 91 days. Feed ingested was recorded daily for each tank, recovering and weighing uneaten feed pellets at the end of each meal. Mortality was also monitored.

### 4.3. Tissue Sampling and Calculation

After a 3-month feeding trial, all fish were subjected to stage three anaesthesia with 300 mg L^−1^ of MSS-222 (Finquel^®^MS-222, Argent Laboratories, Redmont-VI, USA). Biometry measurements (total length, cm and body weight, g) were recorded for the subsequent Fulton’s condition factor calculation [K = fish weight (g) × 100/fish total length (cm)^3^]. After that, three fish in each tank (nine fish/diet) were sacrificed by a lethal dose of the same anesthetic (Finquel^®^MS-222 1 g L^−1^). Blood samples (2 mL) were collected from the caudal vein in S-Monovette^®^ heparin-containing tubes (Sarstedt, Nümbrecht, Germany) and stored in ice and, subsequently, plasma was obtained by blood centrifugation. After the sacrifice, the ventral side of the animals was opened to remove the gastrointestinal tract and the head kidney. The intestinal content, obtained by squeezing the gastrointestinal tract, was collected in sterile cryovials (Eppendorf, USA) and frozen in liquid nitrogen. Subsequently, tissue biopsies from the midgut and head kidney were collected in sterile cryogenic vials CryoTubes^®^ (Merck, Darmstadt, Germany), immediately frozen in liquid nitrogen, and all samples were then stored at −80 °C until assays were performed.

### 4.4. Gut Microbiota Analysis

Total genomic DNA extraction from intestinal contents (90–120 mg) and animal feed was carried out using the DNeasy^®^ PowerSoil^®^ kit (Qiagen, Milan, Italy) following the manufacturer’s instructions but adding Proteinase K after C1 solution, incubating at 70 °C for 10 min and lysing samples in a PowerBead Tube using TissueLyser II (Qiagen, Italy) at maximum power for 10 min. A double elution of the Spin column tube was then made with DEPC-treated water to a final volume of 100 µL. DNA quality and concentration were measured using a NanoDrop 8000 spectrophotometer (Thermo Fisher, Waltham, MA, USA). High-quality DNA was stored at −20 °C.

PCR amplification was performed by targeting 16S rRNA gene V3-V4 variable regions with the bacterial primer set 341F (5′-CCTACGGGNGGCWGCAG-3′) and 806R (5′-GACTACNVGGGTWTCTAATCC-3′), as previously reported [71]. PCR reactions were carried out using 2X KAPA Hifi HotStart Ready Mix (Kapa Biosystems Ltd., UK) according to the following protocol: 5 min at 95 °C, 30 cycles of 30 s at 95 °C, 30 s at 55 °C, and 30 s at 72 °C, followed by a final extension of 5 min at 72 °C. PCR products were checked by gel electrophoresis and cleaned using an Agencourt AMPure XP system (Beckman Coulter, Brea, CA, USA), following the manufacturer’s instructions. After seven PCR cycles (16S Metagenomic Sequencing Library Preparation, Illumina), Illumina adaptors were attached (Illumina Nextera XT Index Primer). Libraries were purified using Agencourt AMPure XP (Beckman) and then sequenced on an Illumina^®^ MiSeq (PE300) platform (MiSeq Control Software 2.0.5 and Real-Time Analysis software 1.16.18, Illumina, San Diego, CA, USA). Sequences obtained from Illumina sequencing were analyzed using the Quantitative Insights Into Microbial Ecology (QIIME) 2.0 pipeline [72]. Unidentified taxa include those whose percentage sequence homology with Greengenes database was less than 95% [73]. α- and β-diversity estimates were determined using the *phyloseq* R Package [74].

### 4.5. Gene Expression in the Midgut and Head Kidney

Total RNA was extracted from 10–30 mg frozen tissue using the TRIzol reagent (Thermo Fisher Scientific, Waltham, MA, USA) according to the manufacturer’s recommendations with the following modifications. Briefly, the tissue was first homogenized in 300 μL Trizol reagent using a sterile plastic pestle, and then 700 μL Trizol reagent was added to a final volume of 1 mL. After homogenization, all steps (including optional steps) were performed as suggested but adding 40 µL of 3 M sodium acetate, 5.5 pH, and 1 mL of cold ethanol 96% to the aqueous phase. Samples were stored overnight at −20 °C and then centrifuged at 13.200× *g* for 30 min at 4 °C. Two washing steps with 500 µL cold 70% ethanol were performed, each followed by 5 min and 6 min centrifugation at 13.000× *g*. Dry pellets were finally resuspended with 50 µL of DEPC-treated water and stored at −80 °C. Extracted total RNA was quantified using a Nanodrop 8000 Spectrophotometer (Thermo Fisher Scientific, Waltham, MA, USA), and RNA quality was assessed using a 2200 TapeStation (Agilent Technologies, Santa Clara, CA, USA). mRNA samples with high quality (RINe > 8) were used for retrotranscription. Reverse transcription was performed with a High-Capacity cDNA Reverse Transcription Kit (Applied Biosystems™, Thermo Fisher Scientific, Waltham, MA, USA) in a 20 µL reaction volume containing 10 µL template RNA (5 ng/µL), 2.0 μL of 10X RT Buffer, 0.8 μL of 25X dNTP Mix (100 mM), 2.0 μL of 10X RT Random Primers, 1.0 μL of MultiScribe™ Reverse Transcriptase, and 4.2 μL of DEPC-treated water. After transcription, cDNA was stored at −20 °C until quantitative Real-Time PCR (RT-PCR).

The expression level of inflammatory genes was determined by RT-PCR using a LightCycler^®^ 480 SW 1.5.1.62 (Roche). RT-PCR was carried out in 20 µL reactions prepared following the manufacturer’s instruction and containing 10 μL of 2x qPCRBIO SyGreen Mix Separate-ROX (PCR BioSystems, UK), 0.4 μL of each primer (10 ng/μL), 5.2 μL of DEPC-treated water, and 4 μL of cDNA (10 ng). Reactions were carried out in triplicate under the following conditions: 95 °C for 15 s, followed by 45 cycles at 95 °C for 15 s, 63 °C for 30 s, 72 °C for 10 s, one cycle at 95 °C for 10 s, 65 °C for 15 s, 97 °C continuous and a final step at 40 °C for 30 s. Ct values for each sample were normalized against the geometric mean Ct values obtained for two housekeeping genes, 18S and EF-1α. Accurate calculation of primer efficiency was evaluated from the standard curve and since it was between 1.96 and 2.0, it was approximated to 100% efficiency for all the target and reference genes. Gene expression was therefore expressed as the relative fold change 2^−ΔΔCt^, where ΔCt was obtained by subtracting the geometric mean Ct for the two reference housekeeping genes 18S and EF-1α from the Ct of the tested gene, and ΔΔCt represented the difference between ΔCt of the test diet compared to the ΔCt of CV as the reference diet. Primers and relative accession numbers are shown in Table 4.

### 4.6. Plasma LPS Concentration

The quantity of 50 µL of plasma was analysed in duplicate. Detection of LPS was performed using an ELISA Kit for Lipopolysaccharide (LPS) (Cloud-Clone Corp, CCC, Katy, TX, USA) according to the manufacturer’s recommendations. Absorbance analysis was carried out using a PowerWave 340 (BioTek) and Gen5™ software.

### 4.7. Statistical Analysis

All statistical analysis was performed using R studio. Percentage relative abundance of taxa from different dietary groups was compared using a nonparametric Wilcoxon statistical test. Normal distribution of data was assessed by Shapiro–Wilk’s test. Pairwise comparison among groups in terms of α-diversity was calculated by the Kruskal–Wallis test, followed by the post-hoc Dunn’s test with Benjamini–Hochberg false discovery rate (FDR) *p* value correction. Differences in the β-diversity were checked using the non-parametric Permutational Multivariate Analysis of Variance (PERMANOVA) and adonis tests with 999 permutations, via the vegan R Package [81]. Correlation between bacterial taxa and experimental ingredients was performed by Spearman’s correlation analysis. After FDR correction, a *p* value < 0.05 was considered statistically significant. All data are expressed as the mean ± standard deviation, SD.

## 5. Conclusions

In summary, our findings suggest that none of the experimental diets negatively affected fish health in terms of growth parameters and inflammatory status. Both poultry by-product and insect meals could be considered valid alternatives to vegetable protein ingredients and may be useful for improving FM-free diets. Changes in bacterial α-diversity after feeding animals with *Hermetia* clearly indicated that a percentage of insect meal in fish feed ranging from 8% to 45% positively modifies the fish gut microbiota. Experimental diets H10, H30, and H60 increased gut bacterial community richness, ameliorating the low diversity profile induced by the vegetable diet and reaching values close to those obtained with the fish meal diet. In addition, according to existing data and to our results, we believe that fermentable chitin should be considered the major driver of positive changes in gut bacterial populations, acting as a prebiotic in trout. However, uptake of insect meal by the aquafeed industry faces cost-effectiveness challenges. For this reason, the combination of insect meal with other alternatives to fish meal such as poultry by-product meal represents a viable alternative in terms of cost, environmental sustainability, and fish health. P addition to VM-based formulations sustained the growth of the animals and maintained a microbiota composition more similar to that of FM-fed animals. In particular, P supported the growth of Proteobacteria, particularly non-pathogenic Gammaproteobacteria, as demonstrated by a comparable relative abundance to that observed in trout fed fish meal. Both *Hermetia* and poultry by-products contributed to increase GM α-diversity, thus rendering it more similar to the GM of fish fed the control fishmeal. GM specifically responded to inclusion of *Hermetia* in vegetable feed with an increase in chitin degraders, i.e., *Actinomyces* and *Bacillus* genera, as well as some Lactobacillales. Similarly, the presence of poultry by-products in the feed promoted the growth of specific proteolytic *Peptostreptococcaceae* and non-pathogenic Proteobacteria. *Weissella* and *Actinomyces* were directly transferred from *Hermetia* feed to trout GM. In other words, the GM of tested animals was able to change dynamically in response to the diet of the host and may play a key role in nutrient digestion.

## Figures and Tables

**Figure 1 ijms-22-05454-f001:**
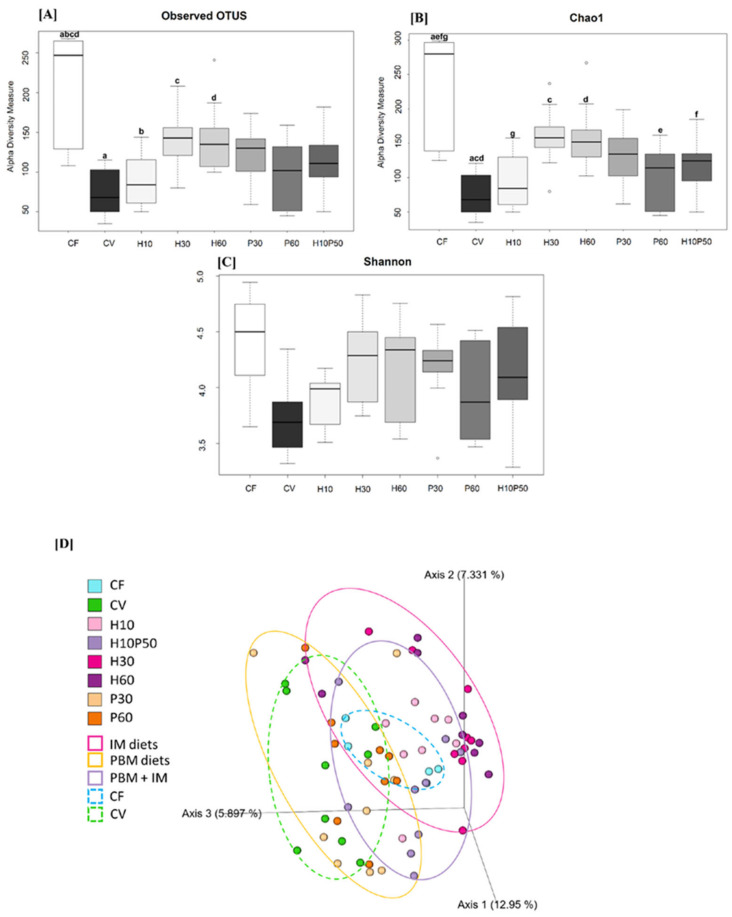
Intestinal bacterial α-diversity (Observed features (**A**), Chao1 (**B**) and Shannon index (**C**)), and β-diversity (Unweighted Unifrac PCoA plot (**D**)) in rainbow trout fed fish-meal (CF), vegetable meal (CV), increasing percentage of *Hermetia illucens* insect meal (H10, H30, H60), increasing percentage of poultry by-product meal (P30, P60), and a combination of insect meal and poultry by product meal (H10P50) (*n* = nine fish per group). Within each panel, identical superscript letters indicate statistically significant differences (FDR *p* < 0.05).

**Figure 2 ijms-22-05454-f002:**
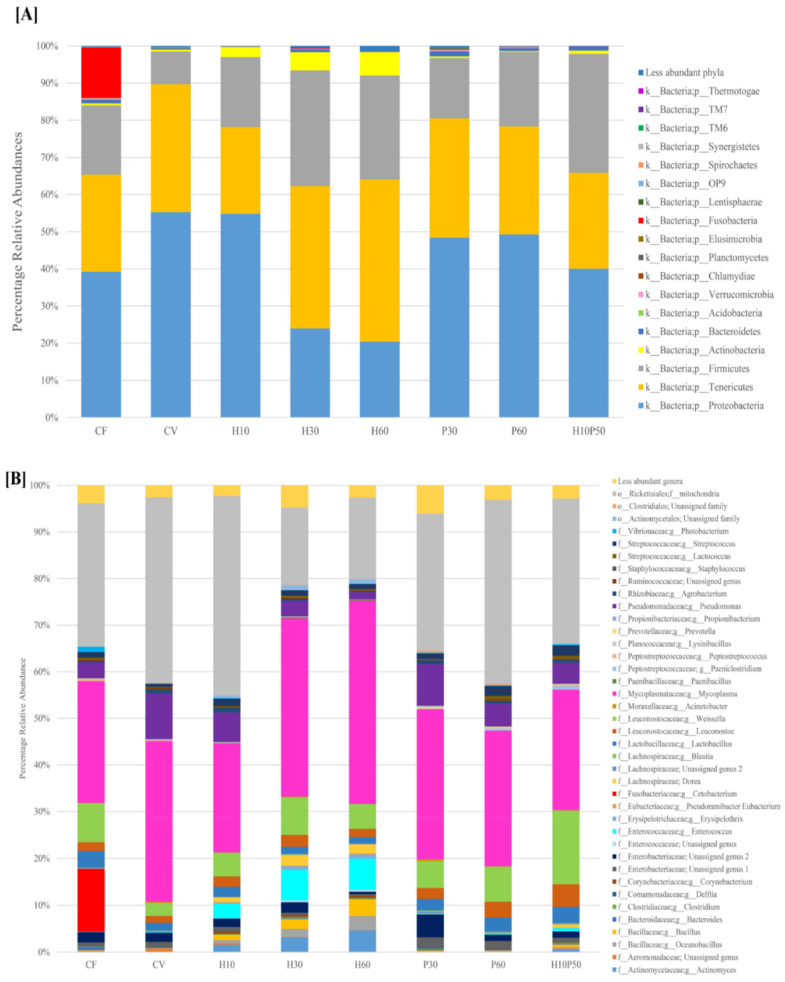
Percentage relative abundance of phyla (**A**) and genera (**B**) in the intestinal contents. Less abundant phyla or genera include bacteria with a relative abundance less than 0.01% in fewer than 25% of samples.

**Figure 3 ijms-22-05454-f003:**
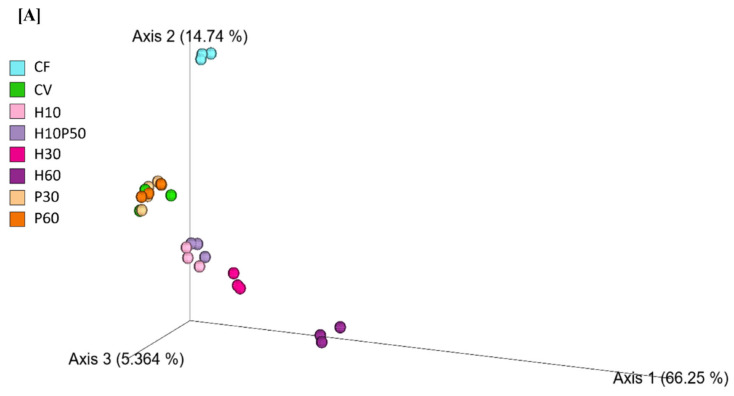
(**A**) PCoA representing the β-diversity of microbial populations in fish feed, according to Bray–Curtis dissimilarity (*p* < 0.001). Percentage relative abundance of phyla (**B**) and genera (**C**) in fish feed. Unassigned phyla or genera include those with percentage sequence homology less than 95% with the Greengenes database. Less abundant phyla or genera include bacteria with a relative abundance less than 0.01% in fewer than 25% of samples.

**Figure 4 ijms-22-05454-f004:**
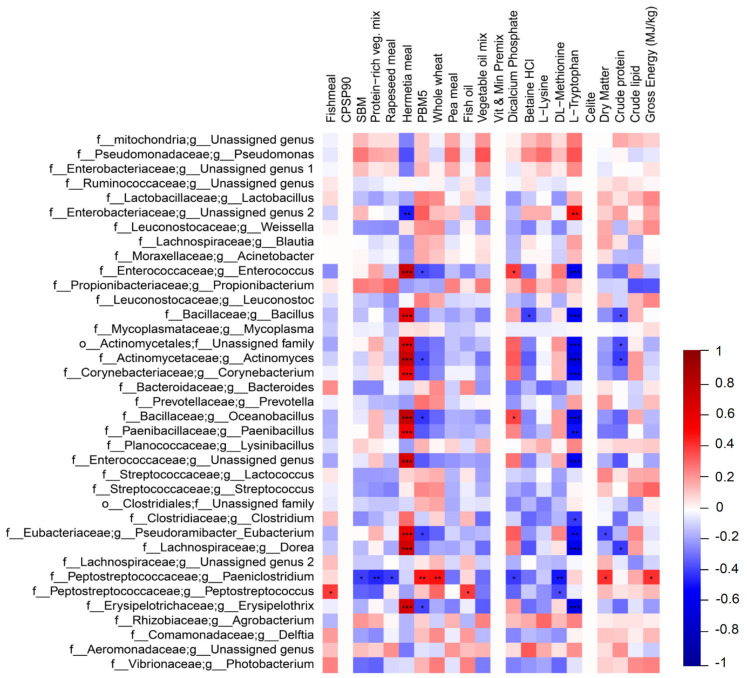
Spearman’s correlation between the relative abundance of gut microbial genera and ingredient composition. A positive correlation is indicated by dark red, a negative correlation by dark blue. Stars indicate statistical significance after FDR correction (* *p* < 0.05, ** *p* < 0.01, *** *p* < 0.001). Families and genera were reported as “Unassigned” when they could not be assigned to any genus (g) or family (f) within the reference database (http://greengenes.lbl.gov, accessed on 13 July 2020), at a percentage sequence homology of 95% or 90% for genus and family, respectively.

**Figure 5 ijms-22-05454-f005:**
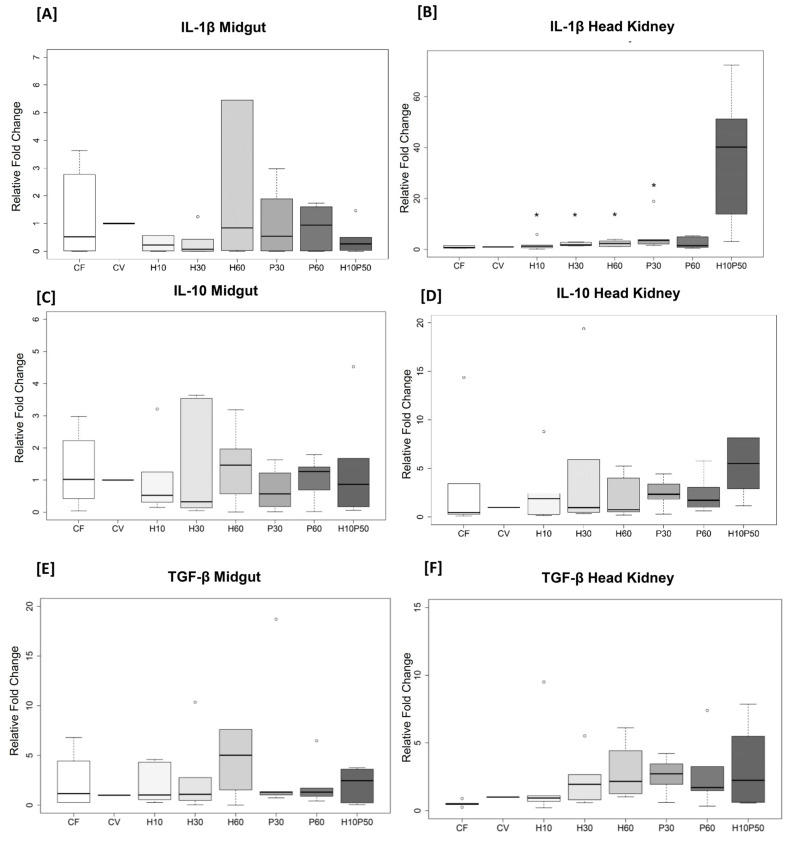
(**A**–**J**). Gene expression of IL-1β, IL-10, TGF-β, COX-2m and TCR-β in midgut and head kidney biopsies (*n* = 6 per diet) expressed as relative fold change 2^−ΔΔCt^ (CV = reference diet). The confidence interval box shows the 95% confidence interval for the median. ° = outliers. * *p* < 0.05 when compared with CV. Note the different scales on the y-axis.

**Table 1 ijms-22-05454-t001:** Final body weight, total length, and Fulton’s condition factor, K = Weight (g) × 100/Total Length (cm^3^) of rainbow trout fed the test diets. Within each column, means with different superscript letters are significantly different (*p* < 0.05).

Diet	Final Weight (g)	Total Length (cm)	K (g/cm^3^)
CF	231.18 ± 32.11 ^b^	25.53 ± 1.35 ^d^	1.39 ± 0.11 ^cd^
CV	227.92 ± 35.22 ^b^	25.01 ± 1.30 ^e^	1.45 ± 0.11 ^a^
H10	235.02 ± 30.41 ^ab^	25.46 ±1.24 ^d^	1.42 ± 0.11 ^abc^
H30	239.1 ± 36.26 ^ab^	25.48 ± 1.42 ^d^	1.44 ± 0.11 ^ab^
H60	241.07 ± 35.58 ^ab^	25.78 ± 1.31 ^c^	1.40 ± 0.10 ^bcd^
P30	240.03 ± 37.82 ^ab^	25.81 ± 1.45 ^bc^	1.39 ± 0.10 ^cd^
P60	244.06 ± 36.02 ^ab^	26.02 ± 1.25 ^ab^	1.38 ± 0.09 ^d^
H10P50	254.81 ± 36.82 ^a^	26.13 ± 1.14 ^a^	1.42 ± 0.12 ^abc^

**Table 2 ijms-22-05454-t002:** (**A**–**B**) Bacterial relative abundances (mean ± SD) of taxonomic groups that were significantly different between diet groups. Identical superscript letters indicate significant differences among treatments according to Benjamini-Hochberg FDR correction. Individual *p* values are reported in the text and in Appendix A.

**[A]**	**Diet**							
	**CF**	**CV**	**H10**	**H30**	**H60**	**P30**	**P60**	**H10P50**
**Phylum**								
Actinobacteria	0.51 ± 0.67	0.52 ± 0.46	2.62 ± 4.26	4.89 ± 4.23	6.29 ± 5.49 ^a^	0.48 ± 0.53	0.23 ± 0.32 ^a^	0.87 ± 0.65
Bacteroidetes	1.00 ± 1.22	0.41 ± 0.69	0.03 ± 0.06	0.72 ± 1.11	0.09 ± 0.09 ^a^	1.35 ± 1.26 ^a^	0.75 ± 0.82	0.77 ± 0.91
Fusobacteria	13.50 ± 29.88 ^a–g^	<0.0001^a^	<0.0001 ^b^	<0.0001^c^	<0.0001 ^d^	<0.0001 ^e^	<0.0001 ^f^	0.002 ± 0.01 ^g^
**Class**								
Erysipelotrichi	0.04 ± 0.06	0.002 ± 0.01 ^ab^	0.34 ± 0.37	0.91 ± 0.80 ^a^	1.11 ± 1.07 ^b^	0.58 ± 0.17	0.04 ± 0.10	0.18 ± 0.31
Fusobacteriia	13.50 ± 29.88 ^a^	<0.0001	<0.0001 ^a^	<0.0001	<0.0001	<0.0001	<0.0001	0.002 ± 0.01
Gammaproteobacteria	7.72 ± 7.05	14.02 ± 14.53^a^	10.76 ± 10.82	6.47 ± 6.36	2.61 ± 2.74 ^a^	17.20 ± 15.41	8.52 ± 5.00	7.59 ± 6.12
Order								
Actinomycetales	0.49 ± 0.66	0.31 ± 0.29	2.50 ± 4.29 ^a^	4.83 ± 4.14 ^b^	6.25 ± 5.46	0.45 ± 0.52	0.20 ± 0.33 ^ab^	0.85 ± 0.65
Bacteroidales	1.00 ± 1.22	0.25 ± 0.55	0.03 ± 0.06	0.69 ± 1.13	0.04 ± 0.05 ^a^	1.14 ± 1.05 ^a^	0.69 ± 0.83	0.47 ± 0.52
Erysipelotrichales	0.04 ± 0.06	0.002 ± 0.001 ^abc^	0.34 ± 0.37 ^a^	0.91 ± 0.80 ^bde^	1.11 ± 1.07 ^cf^	0.06 ± 0.17 ^df^	0.04 ± 0.10 ^e^	0.18 ± 0.31
Fusobacteriales	13.50 ± 29.88 ^abc^	<0.0001	<0.0001 ^a^	<0.0001 ^b^	<0.0001	<0.0001 ^c^	<0.0001 ^d^	0.002 ± 0.01
Pseudomonadales	3.61 ± 2.94	10.08 ± 11.47^a^	7.17 ± 7.51	3.56 ± 3.40	1.54 ± 1.88 ^a^	9.40 ± 8.19	5.06 ± 3.65	4.72 ± 3.73
**Family**								
*Actinomycetaceae*	0.10 ± 0.19	0.05 ± 0.16^abcd^	1.57 ± 2.70 ^aef^	3.16 ± 2.86 ^bgh^	4.66 ± 3.93 ^cil^	0.02 ± 0.06 ^egim^	<0.0001 ^fhln^	0.52 ± 0.40 ^dmn^
*Corynebacteriaceae*	0.10 ± 0.20	0.002 ± 0.007	0.32 ± 0.40 ^a^	0.41 ± 0.53	0.47 ± 0.63	<0.0001 ^a^	0.004 ± 0.01	0.09 ± 0.12
*Bacillaceae*	0.31 ± 0.33	0.35 ± 0.70	1.79 ± 2.73 ^a^	3.83 ± 3.28	6.59 ± 5.60	0.30 ± 0.35	0.18 ± 0.26 ^a^	0.94 ± 0.74
*Paenibacillaceae*	<0.0001	<0.0001 ^a^	0.06 ± 0.11	0.21 ± 0.26	0.20 ± 0.28 ^a^	0.02 ± 0.07	<0.0001	0.02 ± 0.06
*Clostridiaceae*	0.16 ± 0.18	<0.0001 ^a^	0.09 ± 0.23	0.31 ± 0.29	0.24 ± 0.28 ^a^	0.22 ± 0.42	0.26 ± 0.49	0.34 ± 0.62
*Eubacteriaceae*	<0.0001	<0.0001 ^a^	0.16 ± 0.31	0.12 ± 0.18 ^bc^	0.19 ± 0.22 ^a^	<0.0001 ^b^	<0.0001 ^c^	0.02 ± 0.05
*Erysipelotrichaceae*	0.04 ± 0.06	0.002 ± 0.01 ^ab^	0.34 ± 0.37 ^a^	0.91 ± 0.81 ^cd^	1.11 ± 1.07 ^b^	0.06 ± 0.17 ^c^	0.04 ± 0.10 ^d^	0.18 ± 0.31
*Fusobacteriaceae*	13.49 ± 29.88 ^a^	<0.0001	<0.0001 ^a^	<0.0001	<0.0001	<0.0001	<0.0001	<0.0001
**[B]**	**Diet**							
	**CF**	**CV**	**H10**	**H30**	**H60**	**P30**	**P60**	**H10P50**
**Genus**								
*Actinomyces*	0.10 ± 0.18	0.05 ± 0.16 ^abcd^	1.58 ± 2.70 ^aef^	3.16 ± 2.86 ^bgh^	4.66 ± 3.98^cil^	0.02 ± 0.06 ^egi^	<0.0001 ^fhlm^	0.52 ± 0.39 ^dm^
*Oceanobacillus*	0.06 ± 0.14	0.04 ± 0.11 ^ab^	0.63 ± 1.03 ^cd^	1.75 ± 1.54 ^aef^	2.94 ± 2.58 ^bgh^	<0.0001 ^ceg^	<0.0001 ^dfh^	0.35 ± 0.37
*Bacillus*	0.16 ± 0.14	0.07 ± 0.14 ^a^	1.16 ± 1.75	2.08 ± 1.96	3.60 ± 3.09 ^a^	0.25 ± 0.27	0.18 ± 0.26	0.51 ± 0.44
*Pseudoramibacter* *Eubacterium*	<0.0001	<0.0001 ^a^	0.16 ± 0.31	0.12 ± 0.18	0.19 ± 0.22 ^bc^	<0.0001^b^	<0.0001 ^ac^	0.02 ± 0.05
*Dorea*	0.03 ± 0.04	<0.0001 ^abcd^	1.05 ± 1.72 ^ae^	2.26 ± 1.89 ^bf^	1.99 ± 1.71 ^cg^	0.08 ± 0.25	0.09 ± 0.26 ^efgh^	0.71 ± 0.79 ^dh^
*Paeniclostridium*	0.05 ± 0.05	<0.0001 ^a^	0.05 ± 0.13	0.04 ± 0.08	0.02 ± 0.07	0.05 ± 0.10	0.36 ± 0.39	0.77 ± 1.35 ^a^
*Erysipelothrix*	0.04 ± 0.06	<0.0001 ^ab^	0.21 ± 0.41	0.80 ± 0.63 ^acd^	0.89 ± 0.79 ^bef^	<0.0001 ^ce^	0.01 ± 0.03 ^df^	0.13 ± 0.30
*Enterococcus*	0.06 ± 0.09	0.15 ± 0.27 ^ab^	3.09 ± 4.70 ^bfg^	6.48 ± 5.12 ^cdf^	6.68 ± 5.45 ^eg^	0.10 ± 0.21 ^c^	0.17 ± 0.20 ^ade^	0.62 ± 0.81
*Cetobacterium*	13.44 ± 29.91 ^a^	<0.0001	<0.0001 ^a^	<0.0001	<0.0001	<0.0001	<0.0001	<0.0001
*Pseudomonas*	3.48 ± 2.85	9.80 ± 11.06 ^a^	6.36 ± 6.69	3.38 ± 3.10	1.53 ± 1.89 ^a^	8.94 ± 7.85	4.99 ± 3.67	4.56 ± 3.65

**Table 3 ijms-22-05454-t003:** Plasma LPS concentration (ng/μL), mean ± standard deviation (SD).

Diet	LPS Concentration (ng/μL)
CF	240.13 ± 150.13
CV	130.22 ± 111.60
H10	160.23 ± 163.10
H30	117.72 ± 175.90
H60	297.22 ± 347.59
P30	139.84 ± 130.44
P60	176.33 ± 215.96
H10P50	210.21 ± 178.07

**Table 4 ijms-22-05454-t004:** Primers used to evaluate gene expression by RT-PCR.

Gene	Accession Number	Forward Primer (5′-3′)	Reverse Primer (5′-3′)	References
IL-1β	AJ557021	ACATTGCCAACCTCATCATCG	TTGAGCAGGTCCTTGTCCTTG	[75]
IL-10	NM001246350	CGACTTTAAATCTCCCATCGAC	GCATTGGACGATCTCTTTCTT	[76]
COX-2	AJ238307	ATCCTTACTCACTACAAAGG	GCTGGTCCTTTCATGAAGTCTG	[77]
TGF-β	X99303	AGATAAATCGGAGAGTTGCTGTG	CCTGCTCCACCTTGTGTTGT	[78]
TCR-β	AF329700	TCACCAGCAGACTGAGAGTCC	AAGCTGACAATGCAGGTGAATC	[79]
EF-1α	AF498320	ACCCTCCTCTTGGTCGTTTC	TGATGACACCAACAGCAACA	[79]
18S	AF308735	GATCCATTGGAGGGCAAGTCT	CGAGCTTTTTAACTGCAGCAACTTT	[80]

## Data Availability

The data for this study have been deposited in the European Nucleotide Archive (ENA) at EMBL-EBI under accession number PRJEB43152 (https://www.ebi.ac.uk/ena/browser/view/PRJEB43152, accessed on 10 May 2021).

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
