# Peer review of "Processed Animal Proteins from Insect and Poultry By-Products in a Fish Meal-Free Diet for Rainbow Trout: Impact on Intestinal Microbiota and Inflammatory Markers"

_ijms, 2021, doi:10.3390/ijms22115454_

Round 1
Reviewer 1 Report
MDPI
International Journal of Molecular Sciences
IJMS-1217373
In this manuscript, the authors studied the effects of vegetable product protein derivative diet (CV), fish-based diet (CF), insect meal (H) from Hermetia illucens at 10%, 30%, 60%, poultry by-product meal (P) at 30 % and 60%, and combination of H at 10% and P at 50% (H10P50) for 3 months. The authors examined final weight and K value (Table 2), intestinal bacterial α-diversity (Fig. 1), percentage relative abundance of phyla and genera (Fig. 2), bacterial relative abundance of taxonomic groups (Table 3), PCoA representing β-diversity of microbial populations n fish feed, percentage relative abundance of phyla and genera (Fig. 3), spearman’s correlation between relative abundance of gut microbial genera and ingredient (Fig. 4), gene expression of IL-1β, IL-10, TGF-β, Cox-2 and TCR-β in midgut and head kidney (Table 1, Fig. 5), and plasma LPS concentration (Table 4). The authors concluded and reported that H, P and H10P50 could be valid protein sources in fishmeal-free aquafeeds. The authors have provided novel information in the field. Suggest report sources of H, P, CF, CV diets, their chemical compositions, and ingredient for eight diets.
- Table 2: Recheck data, conduct statistical analysis, and make statistical difference among treatments.
- Table 3: Recheck data, conduct statistical analysis, and make statistical difference among treatments.
- Table 4: Recheck data, conduct statistical analysis, and make statistical difference among treatments.
Author Response
The authors thank the reviewer for his/her comments and suggestions, which are specifically addressed as follows.
Reviewer Point 1.1 – Suggest report sources of H, P, CF, CV diets, their chemical compositions, and ingredient for eight diets.
Reply: The ingredient and chemical composition of the eight experimental diets were reported in the original version of the MS as supplementary materials as stated in lines 158-159:
…….”The ingredient composition and proximate analysis of the test diets are shown as supplementary material (Table S1).”
Reviewer Point 1.2 – Table 2, 3 Recheck data, conduct statistical analysis, and make statistical difference among treatments.
Reply: We definitely agree with the comment of Rev1’s and accordingly in Table 2 of the revised version of the MS we have used different letters to indicate statistical differences among dietary treatments. However in order to indicate statistical differences in relative abundance of taxa (Table 3) we use identical superscript letters to indicate significant differences, in order to make it easier for the reader to view the results. We have modified the table legend, specifying that identical superscript letters indicate statistically significant differences. Statistical analysis was performed as described in the Statistical analysis section (lines 249-259) of the original version of the MS.
Reviewer Point 1.3 – Table 4: Recheck data, conduct statistical analysis, and make statistical difference among treatments.
Reply: We have checked again the data and since no significant differences were observed in LPS plasma concentrations among dietary treatments due to the high variability, in the table no letters were used.
In addition the manuscript has been improved in its form, as suggested by the reviewer, and the changes are marked as track changes.
Reviewer 2 Report
This is a very well conducted and described study, and my only major issue is the limited measurement of and discussion of performance metrics. With significant differences detected in overall growth I would have expected the authors to attempt to correlate some of the gut bacterial population structures and gene expression profiles to performance. Were any particular groups of bacteria or gene expressions linked to higher or lower performing feeds. The authors should consider discussing the potential mechanisms that drive the increased performance. Also, was feed consumption recorded? That would be a way to calculate and evaluate either feed conversion ratio or feed efficiency. did some diets perform better or worse simply because the fish ate more or less? and could that be the driver of some of the gut microbiota differences? I also think it would be important to specifically reference how the growth performance metrics align with other FM replacement studies in rainbow trout, as there are many, as well as FM replacement with these same ingredients in any other species as well. This will inform readers as to how these tested diets performed in a much larger comparison and if these feeds are viable or simply had performance differences in this experiment. Finally, just a few specific comments:
Line 58: the (GALT) abbreviation does not fit here, suggest removing until it is actually defined later on line 92.
Line 80: although (H) as an abbreviation for the specific insect meal used was introduced in the abstract, I would suggest redefining it here as it is the first instance in the body of the manuscript
Line 104: suggest moving the scientific name to the first time Rainbow trout is referenced and then only using the common name afterwards
